# Evaluation of collapsible deformation of foundation under rectangular load based on the improved binary medium model

Nadeem Abbas[1], Muhammad Akbar[2,3], S.B.A. Elsayed[4], Gehan Ahmed[5], Ahmed M. Yosri[6]*, Muhammad Usman Arshid[7], Mahmoud Elkady[8]

**1** Department of Disaster Mitigation for Structures, Tongji University, Shanghai, China, **2** School of Naval Architecture & Ocean Engineering, Jiangsu University of Science and Technology, Zhenjiang, Jiangsu, China, **3** Department of Civil Engineering, Northern Border University, Arar, Saudi Arabia, **4** Department of Basic Science, Common First Year Program, Jouf University, Sakaka, Saudi Arabia, **5** Department of Interior Design, College of Engineering, Jouf University, Sakaka, Saudi Arabia, **6** Department of Civil Engineering, College of Engineering, Jouf University, Sakaka, Saudi Arabia, **7** Department of Civil Engineering, University of Engineering and Technology, Taxila, Pakistan, **8** Department of Structural Engineering, Faculty of Engineering, Zagazig University, Zagazig, Egypt

* amyosri@ju.edu.sa

## Abstract

The increasing frequency of extreme weather events and climate change can substantially impact the collapse phenomenon and other challenges associated with the deformation of foundation soils. These can also affect soil moisture regimes, particularly soil suction. The global engineering and geotechnical hazards related to the deformation of foundation soil collapsibility require immediate attention from engineers. The differential equations of the collapsible consolidation deformation of a collapsible loess foundation under concentrated force are formulated using an improved two-dimensional medium model in conjunction with the Biot consolidation theory, fracture mechanics, and continuum theory. The equations are solved using the mathematical and physical methodologies of the Laplace transform and the Hankel transform, and boundary conditions are introduced. The mathematical models of lateral displacement, vertical displacement, and pore water pressure of a collapsible loess foundation with vertical depth, radial distance, and saturation under rectangular load are provided. The proposed model was validated through a series of numerical calculations and analyses. It was demonstrated that the deformation of the collapsible loess foundation under the improved binary medium rectangular load is exceedingly similar to the corresponding engineering deformation. The results of the investigation significantly impact the theoretical research of collapsible loess foundations.

**Data availability statement:** All relevant data are within the paper.

**Funding:** This work was funded by the Deanship of Graduate Studies and Scientific Research at Jouf University under grant No. (DGSSR-2025-02-01012). The funder had no role in study design, data collection and analysis.

**Competing interests:** The authors have declared that no competing interests exist.

# 1. Introduction

The collapsibility of soil foundations poses significant engineering and geotechnical problems globally, whether these soils are naturally occurring or anthropogenically generated, presenting crucial difficulties to engineers [1]. The swift rise in global population has led to urban expansion and the creation of new earthwork infrastructure, making the development of marginal land, which may contain problematic soils like collapsible soils, nearly unavoidable in sustainable construction [2]. Consequently, examining and comprehending the processes behind these events becomes essential. Collapsible loess will collapse rapidly after soaking under certain pressure, which is significantly different from ordinary loess. Collapsible soils show a sudden reduction in volume upon wetting, even without external loading [3]. The Loess Plateaus are widely distributed in the western part of China. Among them, collapsible loess occupies a high proportion, and many buildings are built on the collapsible loess foundation. The uneven collapsible settlement consolidation deformation causes varying degrees of damage to the superstructure, resulting in annual economic losses. In practice, geotechnical engineers face several challenges while working with these. The challenges include (i) the characterization of collapsible soils after their identification. (b) the extent of wetting, (c) the estimation of collapse settlements and strains, and (d) the selection of mitigation option and their design [4,5].

Therefore, the consolidation deformation of collapsible loess foundations is a persistent concern in civil engineering. Qian Hongjin, Wang Jitang, and Luo Yusheng systematically compiled their engineering construction experiences and conducted an analysis of the prevalent issues in collapsible loess regions in China for the first time, alongside extensive experimental research on collapsible loess foundations [6,7]. Hu Changming and Mei Yuan have conducted extensive systematic research on high-fill foundations and slope remediation in collapsible loess regions, examining experimental and numerical dimensions [8]. The collapse potential was shown as a function of the initial void ratio, degree of saturation, thickness of the collapsible layer, soil stress history, and applied load [9,10]. The liquid limit and dry density of soil may signify the potential for collapse in in-situ soil deposits and their geomorphological and geological context [11,12]. They performed a numerical solution, computation, and analysis of the collapsible loess foundation and encapsulated the methodologies and insights [3,13].

With the application of the computer computation method, the development of plastic mechanics, and the advent of fracture mechanics in the last century, the understanding of geotechnical problems has been continuously improved [14]. Many experts and scholars have put forward a series of elastic-plastic constitutive models for the issues of foundation soaking and collapsing deformation, and some of them have been widely used in practical engineering, but all of these existing models have certain limitations [15,16]. In terms of regional, soil, and environmental impacts, especially for soft soil and collapsible loess, no model can solve the collapsibility of loess [17]. At the end of the last century, Shen Zhujiang, Xie defining, and others proposed that the primary research task in the 21st century is to establish the mathematical model of collapsible loess foundation collapsible deformation from the soil structure [18]. Shen Zhujiang has been exploring the constitutive model of collapsible loess

since 1984. In 1994, Shen Zhujiang established the relationship between water content and damage ratio [3,19]. In 1985, he proposed a hyperboloid model, which was applied to solve the problem of soft soil foundations in engineering construction in coastal areas of China. It has gradually improved in the following decades [20].

Meanwhile, the proposed model has been applied to solve the engineering construction problems in the collapsible loess area. Subsequently, various models were proposed on this basis, but none of them could reflect the collapsible deformation of loess [21]. After years of exploration, the binary medium model of loess established in 2002 can basically reflect the characteristics of water immersion and collapsibility of loess foundation, and then it is improved by Chen Tielin, Liu Enlong, and others [22]. A semi-analytical technique for analysing layered saturated clays' creep and thermal consolidation characteristics in response to surface loads [23]. Typical viscoelastic models (e.g., Kelvin, Maxwell, or Merchant), the correspondence principle, and the Laplace-Hankel transform are used to obtain analytical viscoelastic solutions for the long-term behaviour of clays [14]. Several numerical examples are shown to test the theory's validity and investigate the implications of material qualities and stratification on the time-dependent behaviour of thermal consolidation for multilayered transversely isotropic poroelastic material [24]. Two theories validated the correctness of the provided theory, and the impacts of anisotropic permeability and transverse isotropic features on the threnody hydromechanical coupling behaviour of layered saturated media are detailed [25].

Examining the creep and consolidation characteristics of layered saturated soils with superimposed dry layers subjected to vertical loading. A semi-analytical solution is offered for this research using the Laplace–Hankel transform, conventional viscoelastic models (such as the Kelvin, Maxwell, or standard linear solid model), and the correspondence principle [26]. A comprehensive comparison of the current results with existing numerical and analytical findings is provided to validate the solution, accompanied by an in-depth parametric analysis investigating the influence of various viscoelastic models, the thickness of the overlying layer, and viscosity [27].

Cao Jiansheng, Zhang Wanjun, etc. further explored the water volume change law of weathered rock and soil by experimental research under the condition of fully considering the different fillings between rock blocks and fractures [28], Fan Wen, Yan Furong and Lu Quanzhong further verified the adaptability of the binary medium model in loess through the comparative analysis of the calculation results of the binary medium model and the triaxial test results of fractured loess, and established the binary medium model for the mechanical properties of loess in the fractured zone [3], In 2013, Liu Enlong and Zhang Jianhai established a binary medium model of rock under cyclic load through triaxial experiment [29], the developed a triaxial apparatus having controlled suction and obtained the results of tests conducted on undisturbed samples. He reported successful measurement of the collapsible potential of loess sand [30]. Liu Enlong, Hu Zaiqiang, and Hou Feng further explored the applicability of the binary medium model in Loess by using the finite element method through model parameters [3]. However, so far, the error between the calculation results based on the binary medium theory and the actual engineering detection data is still large, especially since the calculation results of horizontal displacement are far less than the actual measurement results, so it has not been effectively applied in real engineering construction in loess area, Therefore, the improve the binary medium. Based on Biot consolidation theory [9], combined with the literature [31], and relevant mathematical and physical methods, the theoretical evolution and collapse deformation of uneven settlement of foundation caused by water immersion are deduced and calculated respectively, and Durbin [30]. Qian H [3] That continuous strip/rectangular footing is safer and more economical than isolated footings on collapsible soils, as it may provide a more economical and safer foundation than isolated footings. Strip footing can effectively control differential settlement by distributing stress more equitably between the isolated columns. Previous studies have primarily focused on evaluating collapsible deformation of the foundation using different methods, such as linear and nonlinear calculations. To deeply understand the deformation behavior of collapsible deformation of foundation composite strata and provide valuable insights into the regional metro station engineering, this paper fills the gap, reviews extensive data, and employs mathematical models to analyze the deformation characteristics of collapsible deformation of foundation. The results provide crucial data for managing collapsible foundation deformation in this region, significantly contributing to

the safety of engineering practices. Table 1 shows the comparisons between the different studies and fills the gap in the literature.

The numerical calculation method of pravin and MATLAB study the collapsible deformation of collapsible loess foundation under rectangular load and compare the theoretical settlement results with the measured values of literature [30].

## 2. Calculation model

Many buildings in collapsible loess areas use independent column foundations. Firstly, the model is simplified, and the independent foundation is simplified into a rectangular load acting on the foundation, as shown in Fig 1.

### 2.1. Deformation theory

In this paper, the binary medium theory is improved, which assumes that:

$$E = M_s / (1 - S_r^n) \qquad (1)$$

$M_s$ Is the initial stress、 $S_r$ Is saturation、 n Is the model parameter, which is indirectly measured by experiment.

The binary medium model regards the collapsible loess foundation as two parts: the structural block and the structural belt. The structural block is an ideal elastic brittle material, the structural belt is a hardened elastic-plastic material, and the structural block and structural belt bears the external load. Namely:

$$\sigma = (1 - b)\,\sigma_i + b\sigma_f \qquad (2)$$

Where: $\sigma_i$ The force on the structural block, $\sigma_f$ The structural belt bears the force, $b$ is the section damage rate.

**Table 1. Author contribution comparisons between the different studies.**

| Authors | Collapsible Loess Type | VDCLF | HDCLF | RCSD | VCT | VCS | Methodology |
|---|---|---|---|---|---|---|---|
| [32] | Deformation Properties of Collapsible Loess Foundation | ✓ | ✓ | x | x | x | Laboratory test |
| [33] | Collapsible loess settling characteristics | x | x | ✓ | ✓ | ✓ | Field immersion test |
| [34] | Evaluation of Settlement Load-Bearing Capacity | ✓ | ✓ | ✓ | x | x | Mathematical Analysis |
| This Study | Collapsible Deformation of Foundation Under Rectangular Load | ✓ | ✓ | ✓ | ✓ | ✓ | Different Mathematical Analysis |

Note: Vertical displacement of collapsible loess foundation, horizontal displacement of collapsible loess, relative collapsibility with saturation depth, collapsibility with time, and collapsibility and saturation.

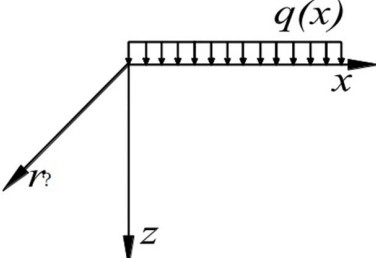

**Fig 1. Calculation Model of semi-infinite collapsible loess foundation.**

According to the two-dimensional medium theory, the load borne by the structural block and the load borne by the structural belt during the collapse of the collapsible loess foundation can be obtained, respectively [3,31]. The load borne by the structural block during the collapsible deformation of collapsible loess foundation is: The load borne by the structure during the collapsible deformation of collapsible loess foundation is: Where: $[D_f]$ Is tangent modulus matrix、 $\{\sigma_f\}$ Does the structural block bear the total stress、 $\{\varepsilon\}$ Is the total strain of the structural belt

## 2.2. Fundamental theory

According to the Prandt-Reuss Law:

$$\left.\begin{array}{l} \sigma_x = B\left[\varepsilon_x - \frac{1}{3}\left(\varepsilon_x + \varepsilon_y + \varepsilon_z\right)\right] + A\frac{1}{3}\left(\varepsilon_x + \varepsilon_y + \varepsilon_z\right) + D \\ \sigma_y = B\left[\varepsilon_y - \frac{1}{3}\left(\varepsilon_x + \varepsilon_y + \varepsilon_z\right)\right] + A\frac{1}{3}\left(\varepsilon_x + \varepsilon_y + \varepsilon_z\right) + D \\ \sigma_z = B\left[\varepsilon_z - \frac{1}{3}\left(\varepsilon_x + \varepsilon_y + \varepsilon_z\right)\right] + A\frac{1}{3}\left(\varepsilon_x + \varepsilon_y + \varepsilon_z\right) + D \\ \tau_{xy} = B\gamma_{xy} + C, \tau_{xz} = B\gamma_{xz} + C, \tau_{zy} = B\gamma_{zy} + C \end{array}\right\}$$

(3)

Where: $A = (1-b)K_i + bK_{ft} - (K_i - K_{fs})\frac{\partial b}{\partial \varepsilon_1}\frac{\partial \varepsilon_1}{\partial \varepsilon_v}$, $B = (1-b)G_i + bG_{ft} - (G_i - G_{fs})\frac{\partial b}{\partial \varepsilon_1}\frac{\partial \varepsilon_1}{\partial \varepsilon_s}$ $D = \frac{\partial K_i}{\partial S_r}\Delta S_r + C$, $C = \frac{\partial G_i}{\partial S_r}\Delta S_r$ and D represents the rate of deformation.

Simplify equation (3) to obtain:

$$\left\{\begin{array}{l} \sigma_x = \left(\frac{2}{3}B + \frac{1}{3}A\right)\varepsilon_x - \left(B - \frac{1}{3}A\right)\varepsilon_y - \left(B - \frac{1}{3}A\right)\varepsilon_z + D \\ \sigma_y = -\left(B - \frac{1}{3}A\right)\varepsilon_x + \left(\frac{2}{3}B + \frac{1}{3}A\right)\varepsilon_y - \left(B - \frac{1}{3}A\right)\varepsilon_z + D \\ \sigma_z = -\left(B - \frac{1}{3}A\right)\varepsilon_x - \left(B - \frac{1}{3}A\right)\varepsilon_y + \left(\frac{2}{3}B + \frac{1}{3}A\right)\varepsilon_z + D \\ \tau_{xy} = B\gamma_{xy} + C, \tau_{xz} = B\gamma_{xz} + C, \tau_{zy} = B\gamma_{zy} + C \end{array}\right.$$

(4)

## 3. Basic equation

### 3.1. Constitutive equation

According to formula (4), the physical equation of collapsible consolidation deformation of collapsible loess foundation under concentrated force is:

$$\left.\begin{array}{l} \sigma_r = a\varepsilon_r + b\varepsilon_\theta + b\varepsilon_z + D + u_w \\ \sigma_\theta = b\varepsilon_r + a\varepsilon_\theta + b\varepsilon_z + D + u_w \\ \sigma_z = b\varepsilon_r + b\varepsilon_\theta + a\varepsilon_z + D + u_w \\ \tau_{rz} = c\gamma_{zr} + C \end{array}\right\}$$

(5)

Where: $a = \frac{2}{3}B + \frac{1}{3}A, b = -\left(B - \frac{1}{3}A\right), c = B$

### 3.2. Geometric equations

$$\varepsilon_r = -\frac{\partial u_r}{\partial r}; \varepsilon_z = -\frac{u_r}{r}; \varepsilon_z = -\frac{\partial w_z}{\partial z}; \gamma_{rz} = -\left(\frac{\partial u_r}{\partial z} + \frac{\partial w_z}{\partial r}\right)$$

(6)

### 3.3. Continuity equation of water

$$-\frac{\partial}{\partial t}\left(\frac{\partial u_r}{\partial r} + \frac{u_r}{r} + \frac{\partial w_z}{\partial z}\right) + k_r\left(\frac{\partial^2}{\partial r^2} + \frac{1}{r}\frac{\partial}{\partial r}\right)u_w + k_z\frac{\partial^2 u_w}{\partial z^2} = 0$$

(7)

Where: $k_r = \frac{k_{rz}}{r_\omega}$, $k_z = \frac{k_{zr}}{r_\omega}$, $k_{rz}$, $k_{zr}$ Is the permeability coefficient in the horizontal and vertical directions, $r_\omega$ Is the unit weight of collapsible loess foundation.

### 3.4. Based on the binary medium model, the collapsible consolidation equation

Bring geometric equation (6) into equation (5):

$$
\begin{cases}
\sigma_r = a\left(-\frac{\partial u_r}{\partial r}\right) + b\left(-\frac{u_r}{r}\right) + b\left(-\frac{\partial w_z}{\partial z}\right) + D + u_w \\
\sigma_\theta = b\left(-\frac{\partial u_r}{\partial r}\right) + a\left(-\frac{u_r}{r}\right) + b\left(-\frac{\partial w_z}{\partial z}\right) + D + u_w \\
\sigma_z = b\left(-\frac{\partial u_r}{\partial r}\right) + b\left(-\frac{u_r}{r}\right) + a\left(-\frac{\partial w_z}{\partial z}\right) + D + u_w \\
\tau_{rz} = -c\left(\frac{\partial u_r}{\partial z} + \frac{\partial w_z}{\partial r}\right) + C
\end{cases}
\tag{8}
$$

Taking equation (8) into equation (5), the calculation simplification and water continuity equation (7) are combined to obtain the consolidation equation of the collapsible loess foundation as follows:

$$
\begin{cases}
a\left(\frac{\partial^2}{\partial r^2} + \frac{1}{r}\frac{\partial}{\partial r} - \frac{1}{r^2}\right)u_r + (b+c)\frac{\partial^2 w_z}{\partial r \partial z} + c\frac{\partial^2 u_r}{\partial z^2} - \frac{\partial D}{\partial r} - \frac{\partial C}{\partial z} - \frac{\partial u_w}{\partial r} = 0 \\
(b+c)\left(\frac{\partial u_r}{\partial r} + \frac{1}{r}\right)\frac{\partial u_r}{\partial z} + a\frac{\partial^2 w_z}{\partial z^2} + c\left(\frac{\partial^2}{\partial r^2} + \frac{1}{r}\frac{\partial}{\partial r}\right)w_z - \frac{\partial D}{\partial z} - \frac{\partial C}{\partial r} - \frac{C}{r} - \frac{\partial u_w}{\partial z} = 0 \\
-\frac{\partial}{\partial t}\left(\frac{\partial u_r}{\partial r} + \frac{u_r}{r} + \frac{\partial w_z}{\partial z}\right) + k_r\left(\frac{\partial^2}{\partial r^2} + \frac{1}{r}\frac{\partial}{\partial r}\right)u_w + k_z\frac{\partial^2 u_w}{\partial z^2} = 0
\end{cases}
\tag{9}
$$

### 3.5. Boundary condition

In this paper, it is assumed that the surface of the collapsible loess foundation is completely permeable, and under the time-varying axisymmetric pressure, the gradient function is used, then:

$$
\left.
\begin{array}{ll}
\sigma(r, z = 0, t) = q\delta(1-r) & 0 \le r \le \infty, 0 \le t \le \infty \\
\tau(r, z = 0, t) = 0 & 0 \le r \le \infty, 0 \le t \le \infty \\
p(r, z = 0, t) = 0 & 0 \le r \le \infty, 0 \le t \le \infty
\end{array}
\right\}
\tag{10}
$$

Where: $\delta(1-r)$ Is a gradient function

### 3.6. Equation solving

By using the differential property of Laplace transform, the consolidation equation (9) of elastic deformation of collapsible loess foundation under concentrated force is transformed with respect to t, that is:

$$
\begin{cases}
a\left(\frac{\partial^2}{\partial r^2} + \frac{1}{r}\frac{\partial}{\partial r} - \frac{1}{r^2}\right)\overline{u}_r + (b+c)\frac{\partial^2 \overline{w}_z}{\partial r \partial z} + c\frac{\partial^2 \overline{u}_r}{\partial z^2} - \frac{\partial \overline{D}}{\partial r} - \frac{\partial \overline{C}}{\partial z} - \frac{\partial \overline{u}_w}{\partial r} = 0 \\
(b+c)\left(\frac{\partial \overline{u}_r}{\partial r} + \frac{1}{r}\right)\frac{\partial \overline{u}_r}{\partial z} + a\frac{\partial^2 \overline{w}_z}{\partial z^2} + c\left(\frac{\partial^2}{\partial r^2} + \frac{1}{r}\frac{\partial}{\partial r}\right)\overline{w}_z - \frac{\partial \overline{D}}{\partial z} - \left(\frac{\partial}{\partial r} - \frac{1}{r}\right)\overline{C} - \frac{\partial \overline{u}_w}{\partial z} = 0 \\
-s\left(\frac{\partial}{\partial r} + \frac{1}{r}\right)\overline{u}_r - s\frac{\partial \overline{w}_z}{\partial z} + k_r\left(\frac{\partial^2}{\partial r^2} + \frac{1}{r}\frac{\partial}{\partial r}\right)\overline{u}_w + k_z\frac{\partial^2 \overline{u}_w}{\partial z^2} = 0
\end{cases}
\tag{11}
$$

In equation (11) $u_r$ Make the Hankel first-order transformation about r, $w_z$, $u_w$ Make the Hankel zero-order transformation of R, that is:

$$
\begin{cases}
-am^2\widetilde{\overline{u}}_r - m(b+c)\frac{\partial \widetilde{\overline{w}}_z}{\partial z} + c\frac{\partial^2 \widetilde{\overline{u}}_r}{\partial z^2} - m\widetilde{\overline{D}} - \frac{\partial \widetilde{\overline{C}}}{\partial z} - m\widetilde{\overline{u}}_w = 0 \\
m(b+c)\frac{\partial \widetilde{\overline{u}}_r}{\partial z} + a\frac{\partial^2 \widetilde{\overline{w}}_z}{\partial z^2} - m^2 c\widetilde{\overline{w}}_z - m\widetilde{\overline{C}} - \frac{\partial \widetilde{\overline{D}}}{\partial z} - \frac{\partial \widetilde{\overline{u}}_w}{\partial z} = 0 \\
sm\widetilde{\overline{u}}_r + s\frac{\partial \widetilde{\overline{w}}_z}{\partial z} + m^2 k_r\widetilde{\overline{u}}_w - k_z\frac{\partial^2 \widetilde{\overline{u}}_w}{\partial z^2} = 0
\end{cases}
\tag{12}
$$

In equation (12) $u_r$, $w_z$, $u_w$ It is a system of second-order non-homogeneous linear differential equations about s and m, and its total solution is the general solution plus special solution, that is:

$$\left\{ \begin{array}{c} \widetilde{\overline{u_r}}(m,z,s) \\ \widetilde{\overline{w_z}}(m,z,s) \\ \widetilde{\overline{u_w}}(m,z,s) \end{array} \right\} = \left\{ \begin{array}{c} \widetilde{\overline{u_{rtoj}}}(m,z,s) \\ \widetilde{\overline{w_{ztoj}}}(m,z,s) \\ \widetilde{\overline{u_{wtoj}}}(m,z,s) \end{array} \right\} + \left\{ \begin{array}{c} \widetilde{\overline{u_{rtej}}}(m,z,s) \\ \widetilde{\overline{w_{ztej}}}(m,z,s) \\ \widetilde{\overline{u_{wtej}}}(m,z,s) \end{array} \right\} \tag{13}$$

The general solution is:

$$\left\{ \begin{array}{c} \widetilde{\overline{u_{rtoj}}}(m,z,s) \\ \widetilde{\overline{w_{ztoj}}}(m,z,s) \\ \widetilde{\overline{u_{wtoj}}}(m,z,s) \end{array} \right\} = \left\{ \begin{array}{c} h_1(m,s) \\ h_2(m,s) \\ h_3(m,s) \end{array} \right\} e^{\lambda_j z} \tag{14}$$

Bring equation (14) into equation (12) and simplify it to obtain:

$$\begin{bmatrix} c\lambda^2 - a\lambda^2 & -m(b+c)\lambda & -m \\ m(b+c)\lambda & a\lambda^2 - m^2c & -\lambda \\ sm & s\lambda & k_r m^2 - k_z \lambda^2 \end{bmatrix} \left\{ \begin{array}{c} h_1 \\ h_2 \\ h_3 \end{array} \right\} = 0 \tag{15}$$

$h_1$, $h_2$, $h_3$ To have a trivial solution, you must have:

$$\lambda^6 + g_1\lambda^4 + g_2\lambda^2 + g_3 = 0 \tag{16}$$

Where:

$$g_1 = \frac{1}{ack_z}\left[ m^2(b+c)^2 k_z - ack_r m^2 - (c^2m^2 + a^2m^2)k_z - sc \right]$$

$$g_2 = \frac{1}{ack_z}\left[ (c^2m^2 + a^2m^2)k_r + acm^4 k_z - m^4(b+c)^2 k_r \right]; g_3 = \frac{1}{ak_z}\left( sm^4 - am^6 k_r \right)$$

Since the inverse Laplace transform is performed in the complex plane s, (16) is treated as a unary sixth-order equation with complex coefficients, and its six roots are expressed as:

$$\lambda_j\,(j=1,2,3) = \left\{ \begin{array}{ll} \lambda_{j1} & x = y = 0 \\ \lambda_{j2} & x = 0 \quad \mathrm{Re}\,[y] > 0 \\ \lambda_{j3} & \text{Other situations} \end{array} \right. \tag{17}$$

$$\lambda_1 = \sqrt{-\frac{g_1}{3}}; \lambda_2 = \sqrt{\left(-\frac{y}{2}-\delta\right)^{\frac{1}{3}}\omega^{j-1} + \frac{-\frac{x}{3}}{\omega^{j-1}\left(-\frac{y}{2}-\delta\right)^{\frac{1}{3}}} - \frac{g_1}{3}};$$

$$\lambda_3 = \sqrt{\left(-\frac{y}{2}+\delta\right)^{\frac{1}{3}}\omega^{j-1} + \frac{-\frac{x}{3}}{\omega^{j-1}\left(-\frac{y}{2}+\delta\right)^{\frac{1}{3}}} - \frac{g_1}{3}}; \lambda_j\,(j=4,5,6) = -\lambda_{j-1} \tag{18}$$

Where: $\delta = \sqrt{\frac{y^2}{4} + \frac{x^3}{27}}$ ($\mathrm{Re}\,[\delta] \geq 0$) Represents a single-valued branch whose real part is positive, $\omega = \frac{-1+\sqrt{3}i}{2}\left(-\frac{y}{2}\pm\delta\right)^{\frac{1}{3}}$ Can take any single-valued branch of cubic radical, $y = \frac{2}{27}g_1^3 - \frac{g_1 g_2}{3} + g_3$ 。 This paper takes $\lambda_j = 1, 2, 3$ Calculate, Take formula (17) and formula (18) into formula (15), and solve it $h_1$, $h_2$, $h_3$ is:

$$h_1 = d\left[ -m\lambda_j^2(b+c) + m\left(a\lambda_j^2 - m^2c\right) + \frac{2\lambda_j^2 m^3(b+c)^2}{c\lambda_j^2 - am^2} \right]; h_2 = d\left[ \lambda_j\left(c\lambda_j^2 - am^2\right) - \lambda_j m^2(b+c) \right]$$

$$h_3 = d\left[ \left(a\lambda_j^2 - m^2c\right)\left(c\lambda_j^2 - am^2\right) + \lambda_j^2 m^2(b+c)^2 \right]$$

Where D is any function about P and S. according to the ordinary differential theory, the solution of equation (14) can be written as:

$$\left\{ \begin{array}{c} \widetilde{\overline{u_{rtoj}}}(m,z,s) \\ \widetilde{\overline{w_{ztoj}}}(m,z,s) \\ \widetilde{\overline{u_{wtoj}}}(m,z,s) \end{array} \right\} = \sum_{j=1}^{3} d_j \left\{ \begin{array}{c} \beta_j(m,s) \\ \alpha_j(m,s) \\ \Delta_j(m,s) \end{array} \right\} e^{\lambda_j z} \tag{19}$$

where: $\beta_j = -m\lambda_j^2(b+c) + m\left(a\lambda_j^2 - m^2 c\right) + \frac{2\lambda_j^2 m^3 (b+c)^2}{c\lambda_j^2 - am^2}; \alpha_j = \lambda_j\left(c\lambda_j^2 - am^2\right) - \lambda_j m^2(b+c); \Delta_j = \left(a\lambda_j^2 - m^2 c\right)\left(c\lambda_j^2 - am^2\right) +$ $d_j(j=1,2,3)$ For 6 undetermined functions with m and s.

For the boundary condition equation (10), the Laplace transform is applied to t, and then the Hankel zero-order transform is applied to R, that is:

$$\left(bmu_{rtoj} + a\frac{\partial w_{ztoj}}{\partial z}\right)_{z=0} = q(t)\frac{J_1(m)}{m}; \left(\frac{\partial u_{rtoj}}{\partial z} - mw_{ztoj}\right)_{z=0} = 0; u_{wtoj}(r,z=0,t) = 0 \tag{20}$$

Bring equation (20) into equation (19):

$$\left. \begin{array}{l} d_1 = qJ_1(m)\left[(\lambda_2\beta_2 - m\alpha_2)\Delta_3 - (\lambda_3\beta_3 - m\alpha_3)\Delta_2\right]e^{\lambda_2 z}e^{\lambda_3 z}/m\Phi \\ d_2 = -qJ_1(m)\left[(\lambda_1\beta_1 - m\alpha_1)\Delta_3 - (\lambda_3\beta_3 - m\alpha_3)\Delta_1\right]e^{\lambda_1 z}e^{\lambda_3 z}/m\Phi \\ d_3 = qJ_1(m)\left[(\lambda_1\beta_1 - m\alpha_1)\Delta_2 - (\lambda_2\beta_2 - m\alpha_2)\Delta_1\right]e^{\lambda_1 z}e^{\lambda_2 z}/m\Phi \end{array} \right\} \tag{21}$$

Where:

$$\begin{aligned} \Phi = & \left(bm\beta_1 e^{\lambda_1 z} + a\lambda_1\alpha_1 e^{\lambda_1 z}\right)\left(\lambda_2\beta_2 e^{\lambda_2 z} - m\alpha_2 e^{\lambda_2 z}\right)\Delta_3 e^{\lambda_3 z} + \left(bm\beta_2 e^{\lambda_2 z} + a\lambda_2\alpha_2 e^{\lambda_2 z}\right)\left(\lambda_3\beta_3 e^{\lambda_3 z} - m\alpha_3 e^{\lambda_3 z}\right)\Delta_1 e^{\lambda_1 z} \\ & + \left(bm\beta_3 e^{\lambda_3 z} + a\lambda_3\alpha_3 e^{\lambda_3 z}\right)\left(\lambda_1\beta_1 e^{\lambda_1 z} - m\alpha_1 e^{\lambda_1 z}\right)\Delta_2 e^{\lambda_2 z} - \left(bm\beta_3 e^{\lambda_3 z} + a\lambda_3\alpha_3 e^{\lambda_3 z}\right)\left(\lambda_2\beta_2 e^{\lambda_2 z} - m\alpha_2 e^{\lambda_2 z}\right)\Delta_1 e^{\lambda_1 z} \\ & - \left(bm\beta_1 e^{\lambda_1 z} + a\lambda_1\alpha_1 e^{\lambda_1 z}\right)\left(\lambda_3\beta_3 e^{\lambda_3 z} - m\alpha_3 e^{\lambda_3 z}\right)\Delta_2 e^{\lambda_2 z} - \left(bm\beta_2 e^{\lambda_2 z} + a\lambda_2\alpha_2 e^{\lambda_2 z}\right)\left(\lambda_1\beta_1 e^{\lambda_1 z} - m\alpha_1 e^{\lambda_1 z}\right)\Delta_3 e^{\lambda_3 z} \end{aligned}$$

According to the form of a solution of a homogeneous higher-order differential equation, the form of a special solution is:

$$\left\{ \begin{array}{c} \widetilde{\overline{u_{rtej}}}(m,z,s) \\ \widetilde{\overline{w_{ztej}}}(m,z,s) \\ \widetilde{\overline{u_{wtej}}}(m,z,s) \end{array} \right\} = \left\{ \begin{array}{c} \psi_1(m,s) \\ \psi_2(m,s) \\ \psi_3(m,s) \end{array} \right\} z^2 e^{\lambda_j z} \tag{22}$$

Bring equation (22) into equation (20) to simplify:

$$\left\{ \begin{array}{l} -\psi_1\left[am^2 M_1 + cM_3\right] - \psi_2 m(b+c)M_2 - \psi_3 mz^2 e^z M_1 = mD \\ \psi_1 m(b+c)M_2 + \psi_2\left[aM_3 - cm^2 M_1\right] - \psi_3 M_2 = mC \\ \psi_1 smz^2 + \psi_2 s\left(2z + \lambda_j z^2\right) + \psi_3\left[m^2 k_r z^2 - k_z\left(2 + 4\lambda_j z + \lambda_j^2 z^2\right)\right] = 0 \end{array} \right. \tag{23}$$

According to Kramer's law, the solution of the equations of equation (23) is:

$$\psi_1(m,s) = \frac{\Theta_1}{\Theta}; \psi_2(m,s) = \frac{\Theta_2}{\Theta}; \psi_3(m,s) = \frac{\Theta_3}{\Theta} \tag{24}$$

$$\Theta_1 = \begin{vmatrix} mD & m(b+c)M_2 & mz^2e^zM_1 \\ mC & aM_3 - cm^2M_1 & M_2 \\ 0 & s(2z + \lambda_j z^2) & m^2k_r z^2 - k_z\left(2 + 4\lambda_j z + \lambda_j^2 z^2\right) \end{vmatrix}$$

$$\Theta_2 = \begin{vmatrix} am^2M_1 + cM_3 & mD & mz^2e^zM_1 \\ m(b+c)M_2 & mC & M_2 \\ smz^2 & 0 & m^2k_r z^2 - k_z\left(2 + 4\lambda_j z + \lambda_j^2 z^2\right) \end{vmatrix}$$

$$\Theta_3 = \begin{vmatrix} am^2M_1 + cM_3 & m(b+c)M_2 & mD \\ m(b+c)M_2 & aM_3 - cm^2M_1 & mC \\ smz^2 & s(2z + \lambda_j z^2) & 0 \end{vmatrix}$$

$$\Theta = \begin{vmatrix} am^2M_1 + cM_3 & m(b+c)M_2 & mz^2e^zM_1 \\ m(b+c)M_2 & aM_3 - cm^2M_1 & M_2 \\ smz^2 & s(2z + \lambda_j z^2) & m^2k_r z^2 - k_z\left(2 + 4\lambda_j z + \lambda_j^2 z^2\right) \end{vmatrix}$$

Bring equation (24) into equation (22), and then bring equation (22) and equation (14) into equation (13). Carry out Laplace Inverse Transformation and Hankel inverse transformation on equation (25), and then obtain the mathematical model of lateral displacement, vertical displacement, and pore water pressure force of elastic deformation of collapsible loess foundation under the action of concentrated force with vertical depth, radial distance, and saturation, namely:

$$u_r(r,z,t) = \frac{1}{2\pi i}\int_{-i\infty}^{i\infty}\int_0^\infty \left(-\sum_{j=1}^3 d_j\beta_j e^{-\lambda_j z} + \psi_1(m,s)z^2 e^{\lambda_j z}\right)mJ_1(mr)e^{st}dmds$$

$$w_z(r,z,t) = \frac{1}{2\pi i}\int_{-i\infty}^{i\infty}\int_0^\infty \left(-\sum_{j=1}^3 d_j\alpha_j e^{-\lambda_j z} + \psi_2(m,s)z^2 e^{\lambda_j z}\right)mJ_0(mr)e^{st}dmds$$

$$u_w(r,z,t) = \frac{1}{2\pi i}\int_{-i\infty}^{i\infty}\int_0^\infty \left(-\sum_{j=1}^3 d_j\Delta_j e^{-\lambda_j z} + \psi_3(m,s)z^2 e^{\lambda_j z}\right)mJ_0(mr)e^{st}dmds$$

$$(25)$$

Using the same solution method, through equation degradation, the mathematical expressions of the corresponding transverse displacement, vertical displacement, and pore water pressure with depth, radial distance, and saturation during elastic deformation and elastic-plastic deformation of collapsible loess foundation under concentrated force can be obtained, namely:

$$u_r(r,z,t) = \frac{1}{2\pi i}\int_{-i\infty}^{i\infty}\int_0^\infty -\sum_{j=1}^3 d_j\beta_j e^{-\lambda_j z} mJ_0(mr)e^{st}dmds$$

$$w_z(r,z,t) = \frac{1}{2\pi i}\int_{-i\infty}^{i\infty}\int_0^\infty -\sum_{j=1}^3 d_j\alpha_j e^{-\lambda_j z} mJ_0(mr)e^{st}dmds$$

$$u_w(r,z,t) = \frac{1}{2\pi i}\int_{-i\infty}^{i\infty}\int_0^\infty -\sum_{j=1}^3 d_j\Delta_j e^{-\lambda_j z} mJ_0(mr)e^{st}dmds$$

Although the improved binary medium model provides a valuable theoretical framework for evaluating the collapsible deformation of foundations under rectangular loads, its assumptions of foundation homogeneity, immediate saturation, and neglect of time-dependent effects may limit its direct applicability in complex field conditions. Therefore, further calibration with experimental or theoretical data is recommended for practical engineering applications. This model improves upon classical elastic or plastic models by considering soil as a binary medium. It combines a solid skeleton and pore structure behavior, which is especially important for collapsible loess foundations.

## 4. Analysis of the improved binary medium model

### 4.1. Model validation

To validate the collapsible consolidation deformation model of a collapsible loess foundation, we will use the enhanced binary medium, collapsible consolidation deformation model collapsible loess foundation based on the improved binary medium, the relative vertical displacement and collapsible deformation of the collapsible consolidation deformation model of collapsible loess foundation based on the improved binary medium are compared with the measured values in the literature [17]. (If the values of [3], are experimental results, through comparison, it is found that the calculated values are well in compliance with actually measured values; Fig 2 shows the calculated and measured relative collapsibility (mm), and it returns an excellent coefficient of determination between the measured and calculated values as the changing trend is consistent.

### 4.2. Comparative analysis of the vertical displacement of collapsible loess foundation

Fig 3 illustrates the results of calculating the vertical displacement of the collapsible loess foundation. The influence depth of the improved binary medium model on vertical displacement exceeds that of the linear consolidation deformation model by 20%. According to the improved binary medium model, the vertical displacement of the collapsible loess foundation is 33 cm, while the nonlinear consolidation deformation model yields a displacement of 29 cm, and the linear consolidation deformation model results in a displacement of 24 cm. The mathematical comparison of the various models confirms the accuracy of the improved binary medium model's results.

Fig 4 presents the results of calculating the horizontal displacement of the collapsible loess foundation. The horizontal displacement value in the binary medium model is lower than the results obtained from the linear and nonlinear consolidation deformation models. In the improved binary medium model, the horizontal displacement of the collapsible loess foundation is 18 cm, while the nonlinear consolidation deformation model yields a displacement of 21 cm, and the linear consolidation deformation model results in a displacement of 25 cm.

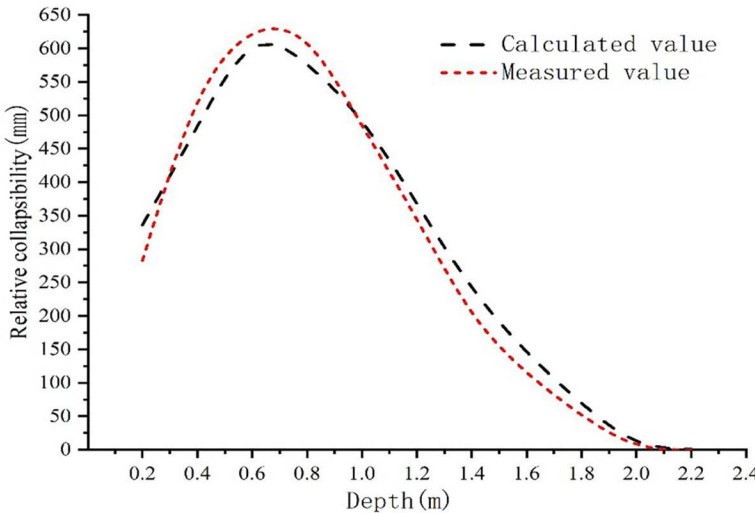

**Fig 2. Comparison between the calculated value and the theoretical value.**

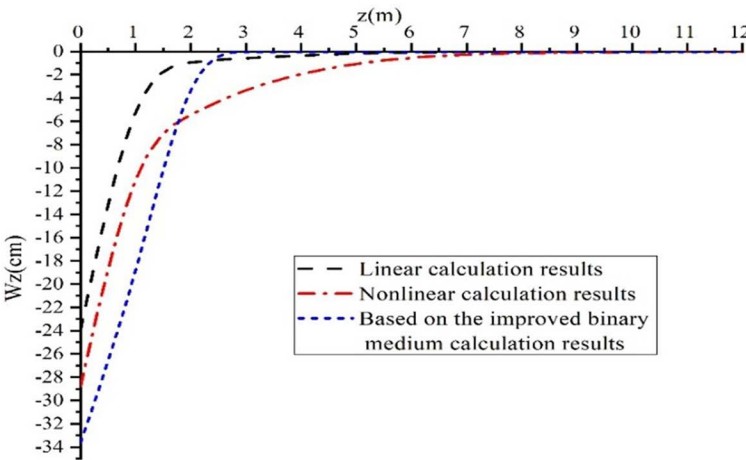

**Fig 3. Comparative Analysis of the vertical displacement of collapsible loess foundation.**

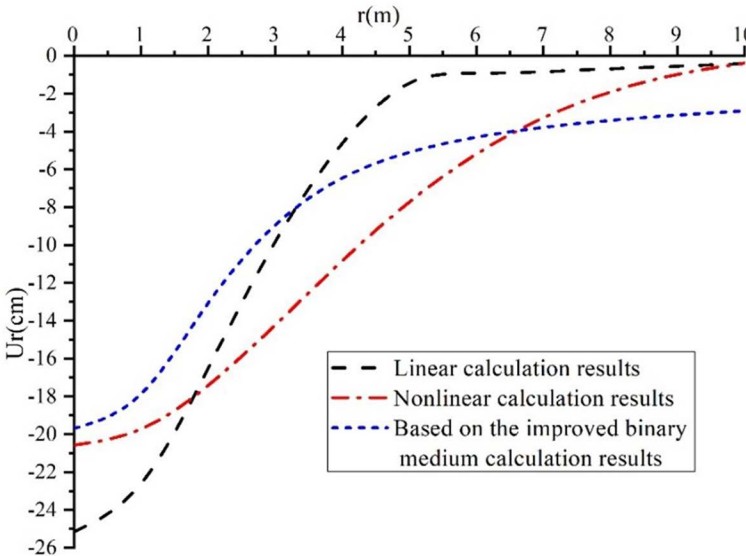

**Fig 4. Comparative analysis of horizontal displacement of collapsible loess.**

### 4.3. Variation of relative collapsibility with saturation depth

Fig 5 displays the results of the calculations and analyses that were performed on the variation relationship of vertical displacement relative to collapsibility with depth under various saturation conditions. As determined by computation, the relative collapsibility of vertical displacement rises with the increase in saturation (Sr) at the same depth. At the same depth, the change in the relative collapsibility of vertical displacement is relatively close, specifically between saturation 0 and 0.2. However, as the saturation increases, specifically between 0.2 and 1, the relative collapsibility of vertical displacement changes more quickly. This is especially true when the saturation is greater than 0.4. Furthermore, it has been discovered that the maximal value of relative collapse of vertical displacement rises with the rise in depth, and this phenomenon occurs in conjunction with an increase in saturation.

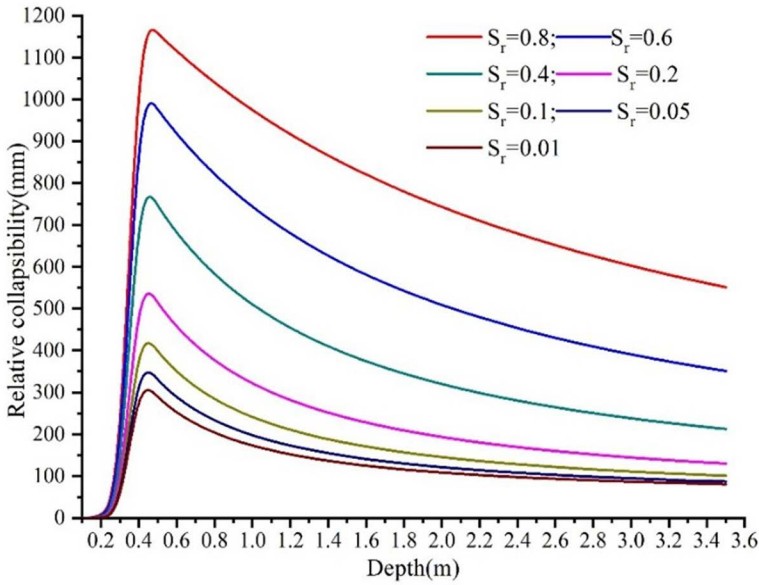

**Fig 5. Variation of relative collapsibility with saturation depth.**

## 4.4. Variation of collapsibility with time and between collapsibility and saturation

Calculations and analyses have shown that the relative collapsibility of vertical displacement steadily decreases with increasing depth, provided that the saturation level remains the same at the same time. With the passage of time, the relative collapsibility of vertical displacement eventually tends to a stable value, even when the saturation level remains the same and the depths vary. At the same time, it has been discovered that the relative collapse deformation is faster, the relative collapse deformation gradually decreases with time in approximately two to eight hours, and the relative collapse

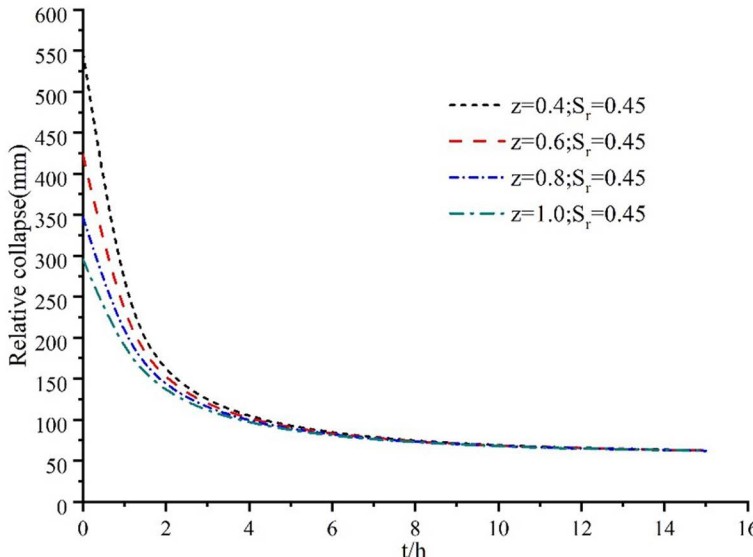

**Fig 6. Variation of collapsibility with time.**

deformation gradually begins to stabilize after eight hours, as depicted in Fig 6. This is because the relative collapse amount of vertical displacement at different depths of the same saturation increases with time in the first two hours.

While this is going on, the variation in the collapsibility of vertical displacement with saturation at various depths is also being investigated, as can be seen in Fig 7. At the same saturation level, the collapsibility of vertical displacement reduces with increasing depth. This is something that can be determined by computation and study. The collapsibility of vertical displacement decreases with increasing saturation as well. Under conditions where the saturation is less than 0.4, the saturation rises and changes rapidly, but under conditions where the saturation is larger than 0.4, the saturation grows and changes slowly.

## 5. Conclusion

Based on the improved two-dimensional medium model, combined with Biot consolidation theory, fracture mechanics, and continuum theory, the differential equations of collapsible consolidation deformation of collapsible loess foundation under concentrated force are established, the boundary conditions are introduced, and the mathematical and physical methods of Laplace transform and Hankel transform are used to solve the equations. The mathematical models of lateral displacement, vertical displacement, and pore water pressure of collapsible consolidation deformation of collapsible loess foundation under rectangular load with vertical depth, radial distance, and saturation are given. At the same time, the model is analyzed; through the calculation and analysis, the following main conclusions are drawn.

- At different saturations, the relative collapsibility of vertical displacement first increases and then decreases with the depth increase. It increases rapidly to the peak value with depth and then decreases gradually.

- At the same depth, the change of relative collapsibility of vertical displacement is basically close between saturation 0~0.2. Between saturation 0.2~1, with the increase of saturation, the relative collapsibility of vertical displacement changes faster, especially after saturation is greater than 0.4.

- The collapsibility of vertical displacement decreases with the increase of depth at the same saturation, and at the same depth, it decreases with the higher saturation. When the saturation is less than 0.4, it increases and changes quickly; when it is greater than 0.4, it increases and changes slowly.

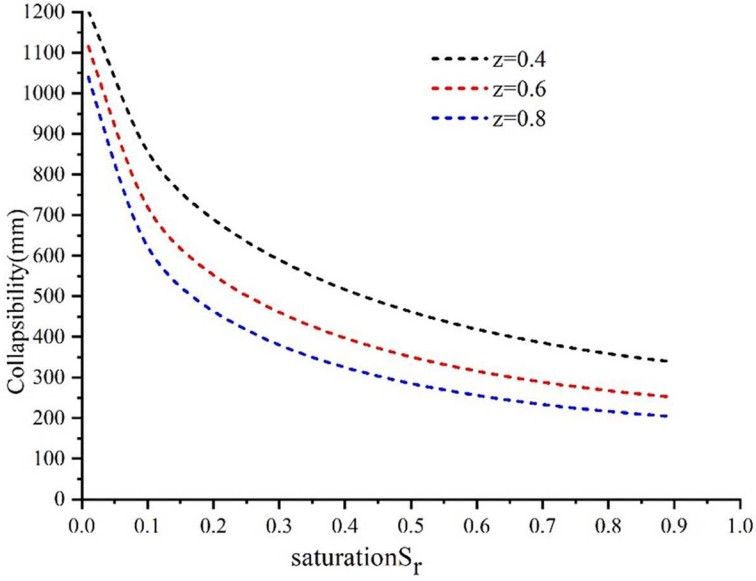

**Fig 7. The variation relationship between collapsibility and saturation.**

- The relative collapsibility of vertical displacement at different depths of the same saturation increases with time in the first 2h, the relative collapsibility deformation decreases with time in about 2h~8h, and the relative collapsibility deformation begins to stabilize after 8h.

**Nomenclate**

| Name | Symbol |
|---|---|
| Initial Stress | $M_s$ |
| Saturation | $S_r$ |
| Model Parameter | n |
| Force on the structural block | $\sigma_i$ |
| The structural belt bears the force | $\sigma_f$ |
| Section damage rate | $b$ |
| Tangent modulus matrix | $[D_f]$ |
| The structural block bear the total stress | $\{\sigma_f\}$ |
| Total strain of the structural belt | $\{\varepsilon\}$ |
| Rate of deformation | D |
| Coefficient of permeability | $K$ |
| Gradient function | $\delta(1-r)$ |
| Time | t |
| Correction coefficient of collapsibility | $\beta$ |
| Collapsible coefficient | $\delta$ |
| Specific gravity | $G$ |
| Hankel first-order transformation | $u_r$ |
| Hankel zero-order transformation | $w_z$, $u_w$ |

## Author contributions

**Conceptualization:** Nadeem Abbas, Ahmed M. Yosri.

**Data curation:** Nadeem Abbas, Muhammad Akbar.

**Formal analysis:** Muhammad Akbar.

**Investigation:** Muhammad Usman Arshid.

**Methodology:** Muhammad Akbar, Ahmed M. Yosri, Muhammad Usman Arshid, Mahmoud Elkady.

**Resources:** Nadeem Abbas, Muhammad Akbar, Muhammad Usman Arshid.

**Supervision:** Gehan Ahmed.

**Validation:** S.B.A. Elsayed, Gehan Ahmed, Muhammad Usman Arshid, Mahmoud Elkady.

**Visualization:** Gehan Ahmed, Muhammad Usman Arshid.

**Writing – original draft:** S.B.A. Elsayed, Gehan Ahmed, Mahmoud Elkady.

**Writing – review & editing:** S.B.A. Elsayed, Ahmed M. Yosri, Mahmoud Elkady.

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
