## [Decision Letter · Decision Letter 0]

2 Apr 2025

PONE-D-25-11186Evaluation of Collapsible Deformation of foundation under rectangular load based on the improved binary medium modelPLOS ONE

Dear Dr. Yosri,

Thank you for submitting your manuscript to PLOS ONE. After careful consideration, we feel that it has merit but does not fully meet PLOS ONE’s publication criteria as it currently stands. Therefore, we invite you to submit a revised version of the manuscript that addresses the points raised during the review process.

**ACADEMIC EDITOR: **

Please pay attentions to the different mechanisms and time scales of collapse and consolidation.  ==============================Please submit your revised manuscript by May 17 2025 11:59PM. If you will need more time than this to complete your revisions, please reply to this message or contact the journal office at plosone@plos.org . Please include the following items when submitting your revised manuscript:

We look forward to receiving your revised manuscript.

Kind regards,

Jianguo Wang, PhD

Academic Editor

PLOS ONE

“Deanship of Graduate Studies and Scientific Research at Jouf University under grant No. (DGSSR-2023-02-02040).”

“The authors extend their appreciation to the Deanship of Graduate Studies and Scientific Research at Jouf University for funding this work under grant No. (DGSSR-2023-02-02040).”

“Deanship of Graduate Studies and Scientific Research at Jouf University under grant No. (DGSSR-2023-02-02040).”

Reviewers' comments:

Reviewer's Responses to Questions

**Comments to the Author**

1. Is the manuscript technically sound, and do the data support the conclusions?

Reviewer #1: Yes

2. Has the statistical analysis been performed appropriately and rigorously? 

Reviewer #1: No

3. Have the authors made all data underlying the findings in their manuscript fully available?

Reviewer #1: Yes

4. Is the manuscript presented in an intelligible fashion and written in standard English?

Reviewer #1: No

5. Review Comments to the Author

Reviewer #1: Collapsible loess foundation will collapse rapidly after soaking under certain pressure, whose consolidation deformation is significantly different from ordinary loess. With the aid of the Laplace–Hankel transform, differential equations of collapsible consolidation deformation of collapsible loess foundation under concentrated force are solved. A series of numerical examples are conducted to investigate the relative collapsibility and horizontal displacement of collapsible loess foundation. In general, this manuscript can provide a reference for the design of collapsible loess foundation under rectangular load. The manuscript needs major revisions. Here are the reviewer’s comments:

1. There are two sentence tenses in the Abstract, please unify them.

2. Some of the words in the manuscript are wrong. For example, “the threnody hydromechanical coupling behaviour of layered saturated media” in Introduction should be revised. Page and line numbers are missing

3. In Eq.(3), the meaning of the symbols that appear in the equation should be explained, especially the symbols that define the soil properties.

4 The construction process can be omitted, and similar steps are already in place in published papers.

5. The assumptions and limitations of the presented model should be added.

6. While improved two-dimensional medium model outperforms traditional models in deformation of collapsible loess foundation, the specific mathematical or physical mechanisms of improvement are unclear. A detailed comparative analysis with classical models should be added.

7. For the calculations and analysis, practical control measures for horizontal deformation or relative collapsibility are missing. It is not enough to analyze the increase and decrease of the curve.

6. PLOS authors have the option to publish the peer review history of their article (what does this mean? ). If published, this will include your full peer review and any attached files.

**Do you want your identity to be public for this peer review?** For information about this choice, including consent withdrawal, please see our Privacy Policy .

Reviewer #1: No

---

## [Author Response · Author response to Decision Letter 1]

13 May 2025

Changes made to the Paper as Major Revision

Paper Title: “Evaluation of Collapsible Deformation of foundation under rectangular load based on the improved binary medium model”

Dear Editor

The authors gratefully acknowledge the constructive comments on the paper entitled “Evaluation of Collapsible Deformation of foundation under rectangular load based on the improved binary medium model” offered by the editor and the anonymous referees. We have reviewed the comments and generally agree with those. Accordingly, we have made changes in the paper in line with the referees’ observations to the extent practicable. We take this opportunity to thank each of the individuals involved in the process. We express our sincere gratitude for their in-depth reviews, which have helped us significantly improve the quality of the paper.

Briefly, the following changes were made:

1. The manuscript has been revised extensively throughout to improve readability.

2. Inconsistent things have been removed from the revised manuscript.

We have addressed all the reviewers’ comments and made significant revisions to the manuscript, which strengthen it considerably from the original submission.

Please find below our response to each of the comments in the order in which they were raised.

Regards,

Corresponding author

Academic Editor

Academic Editor Comments: Please pay attention to the different mechanisms and time scales of collapse and consolidation.

Response:

I would like to thank the Academic Editor. The author has revised the manuscript according to the editor's suggestions.

Variation of relative collapsibility with saturation depth

Figure 5 displays the results of the calculations and analyses that were performed on the variation relationship of vertical displacement relative to collapsibility with depth under various saturation conditions. As determined by computation, the relative collapsibility of vertical displacement rises with the increase in saturation (Sr) at the same depth. At the same depth, the change in the relative collapsibility of vertical displacement is relatively close, specifically between saturation 0 and 0.2. However, as the saturation increases, specifically between 0.2 and 1, the relative collapsibility of vertical displacement changes more quickly. This is especially true when the saturation is greater than 0.4. Furthermore, it has been discovered that the maximal value of relative collapse of vertical displacement rises with the rise in depth, and this phenomenon occurs in conjunction with an increase in saturation.

Figure 5. Variation of relative collapsibility with saturation depth

Variation of collapsibility with time and between collapsibility and saturation

Calculations and analyses have shown that the relative collapsibility of vertical displacement steadily decreases with increasing depth, provided that the saturation level remains the same at the same time. With the passage of time, the relative collapsibility of vertical displacement eventually tends to a stable value, even when the saturation level remains the same and the depths vary. At the same time, it has been discovered that the relative collapse deformation is faster, the relative collapse deformation gradually decreases with time in approximately two to eight hours, and the relative collapse deformation gradually begins to stabilize after eight hours, as depicted in Figure 6. This is because the relative collapse amount of vertical displacement at different depths of the same saturation increases with time in the first two hours.

Figure 6. Variation of collapsibility with time

While this is going on, the variation in the collapsibility of vertical displacement with saturation at various depths is also being investigated, as can be seen in Figure 7. At the same saturation level, the collapsibility of vertical displacement reduces with increasing depth. This is something that can be determined by computation and study. The collapsibility of vertical displacement decreases with increasing saturation as well. Under conditions where the saturation is less than 0.4, the saturation rises and changes rapidly, but under conditions where the saturation is larger than 0.4, the saturation grows and changes slowly.

Figure 7. The variation relationship between collapsibility and saturation

Changes made to the Paper as Major Revision

Paper Title: “Evaluation of Collapsible Deformation of foundation under rectangular load based on the improved binary medium model”

Dear Editor

The authors gratefully acknowledge the constructive comments on the paper entitled “Evaluation of Collapsible Deformation of foundation under rectangular load based on the improved binary medium model” offered by the editor and the anonymous referees. We have reviewed the comments and generally agree with those. Accordingly, we have made changes in the paper in line with the referees’ observations to the extent practicable. We take this opportunity to thank each of the individuals involved in the process. We express our sincere gratitude for their in-depth reviews, which have helped us significantly improve the quality of the paper.

Briefly, the following changes were made:

3. The manuscript has been revised extensively throughout to improve readability.

4. Inconsistent things have been removed from the revised manuscript.

We have addressed all the reviewers’ comments and made significant revisions to the manuscript, which strengthen it considerably from the original submission.

Please find below our response to each of the comments in the order in which they were raised.

Regards,

Corresponding author

Reviewer 1 Comments

A collapsible loess foundation will collapse rapidly after soaking under certain pressure, whose consolidation deformation is significantly different from ordinary loess. With the aid of the Laplace–Hankel transform, differential equations of collapsible consolidation deformation of collapsible loess foundation under concentrated force are solved. A series of numerical examples are conducted to investigate the relative collapsibility and horizontal displacement of collapsible loess foundation. In general, this manuscript can provide a reference for the design of collapsible loess foundation under rectangular load. The manuscript needs major revisions. Here are the reviewer’s comments:

Reviewer Comments: There are two sentence tenses in the Abstract, please unify them.

Response:

I would like to express my sincere gratitude to the esteemed reviewers for their valuable comments and suggestions. The abstract manuscript has been revised accordingly and aligns with the reviewers' feedback.

The increasing frequency of extreme weather events and climate change can substantially impact the collapse phenomenon and other challenges associated with the deformation of foundation soils. These can also affect soil moisture regimes, particularly soil suction. The global engineering and geotechnical hazards related to the deformation of foundation soil collapsibility require immediate attention from engineers. The differential equations of the collapsible consolidation deformation of a collapsible loess foundation under concentrated force are formulated using an improved two-dimensional medium model in conjunction with the Biot consolidation theory, fracture mechanics, and continuum theory. The equations are solved using the mathematical and physical methodologies of Laplace transform and Hankel transform, and boundary conditions are introduced. The mathematical models of lateral displacement, vertical displacement, and pore water pressure of a collapsible loess foundation with vertical depth, radial distance, and saturation under rectangular load are provided. The proposed model was validated through a series of numerical calculations and analyses. It was demonstrated that the deformation of the collapsible loess foundation under the improved binary medium rectangular load is exceedingly similar to the corresponding engineering deformation. The results of the investigation significantly impact the theoretical research of collapsible loess foundations.

Reviewer Comments: Some of the words in the manuscript are wrong. For example, “the threnody hydromechanical coupling behaviour of layered saturated media” in the Introduction should be revised. Page and line numbers are missing

Response:

I would like to express my sincere gratitude to the esteemed reviewers for their valuable comments and suggestions. The manuscript's introduction has been revised accordingly and aligns with the reviewers' feedback.

The collapsibility of soil foundations poses significant engineering and geotechnical problems globally, whether these soils are naturally occurring or anthropogenically generated, presenting crucial difficulties to engineers (Fatahizadeh M et al., 2024). The swift rise in global population has led to urban expansion and the creation of new earthwork infrastructure, making the development of marginal land, which may contain problematic soils like collapsible soils, nearly unavoidable in sustainable construction (Mahmood, M.S et al., 2021). Consequently, examining and comprehending the processes behind these events becomes essential. Collapsible loess will collapse rapidly after soaking under certain pressure, which is significantly different from ordinary loess. Collapsible soils show a sudden reduction in volume upon wetting, even without external loading (Qian, H., Wang, J. and Luo, Y., 1985). The Loess Plateaus are widely distributed in the western part of China. Among them, collapsible loess occupies a high proportion, and many buildings are built on the collapsible loess foundation. The uneven collapsible settlement consolidation deformation causes varying degrees of damage to the superstructure, resulting in annual economic losses. In practice, geotechnical engineers face several challenges while working with these. The challenges include (i) the characterization of collapsible soils after their identification. (b) the extent of wetting, (c) the estimation of collapse settlements and strains, and (d) the selection of mitigation option and their design (Jian, G. and Wen, S., 2022; Luo Yusheng., 2008).

Therefore, the consolidation deformation of collapsible loess foundations is a constant focus in civil engineering. Qian Hongjin, Wang Jitang and Luo Yusheng systematically summarized their engineering construction experience and analyzed existing problems in collapsible loess areas in China for the first time and did a lot of experimental research on collapsible loess foundations (Cheng, H et al., 2016; Yenes M et al., 2012). Hu Changming and Mei Yuan have done a lot of systematic research on high-fill foundation and slope treatment in collapsible loess areas from experimental and numerical aspects (Ozer M et al., 2012). They presented the collapse potential as a function of the initial void ratio, degree of saturation, thickness of the collapsible stratum, stress history of the soil, and the applied load [29,30]. The liquid limit and dry density of soil may indicate the collapsible potential of the in-situ soil deposits and their geomorphological and geological setting (Abbas, A. et al., 2024; Zhang, Y. et al., 2023). They conducted a numerical solution, calculation, and analysis of the collapsible loess foundation and summarized the experience and methods (Nadeem M. et al., 2021; Qian H. et al., 1985).

With the application of the computer computation method, the development of plastic mechanics, and the advent of fracture mechanics in the last century, the understanding of geotechnical problems has been continuously improved (Li, H.R. et al., 2012). Many experts and scholars have put forward a series of elastic-plastic constitutive models for the issues of foundation soaking and collapsing deformation, and some of them have been widely used in practical engineering, but all of these existing models have certain limitations (Azam, S., 2000). In terms of regional, soil, and environmental impacts, especially for soft soil and collapsible loess, no model can solve the collapsibility of loess (Shen, Z.J., 2005). At the end of the last century, Shen Zhujiang, Xie defining, and others proposed that the primary research task in the 21st century is to establish the mathematical model of collapsible loess foundation collapsible deformation from the soil structure (Shi, X.S et al., 2012). Shen Zhujiang has been exploring the constitutive model of collapsible loess since 1984. 1994 Shen Zhujiang established the relationship between water content and damage ratio (Qian, H et al.,1985; Biot, M.A, 1941). In 1985, he proposed a hyperboloid model, which was applied to solve the problem of soft soil foundations in engineering construction in coastal areas of China. It has gradually improved in the following decades (Reznik, Y.M., 2007).

Meanwhile, the proposed model has been applied to solve the engineering construction problems in the collapsible loess area. Subsequently, various models were proposed on this basis, but none of them could reflect the collapsible deformation of loess (Gao, X. et al., 2020). After years of exploration, the binary medium model of loess established in 2002 can basically reflect the characteristics of water immersion and collapsibility of loess foundation, and then it is improved by Chen Tielin, Liu Enlong and others (Wen, F.A.N. et al., 2009). A semi-analytical technique for analysing layered saturated clays' creep and thermal consolidation characteristics in response to surface loads (Fan, W. et al., 2018). Typical viscoelastic models (e.g., Kelvin, Maxwell, or Merchant), the correspondence principle, and the Laplace-Hankel transform are used to obtain analytical viscoelastic solutions for the long-term behaviour of clays (Li, H.R. et al., 2012). Several numerical examples are shown to test the theory's validity and investigate the implications of material qualities and stratification on the time-dependent behaviour of thermal consolidation for multilayered transversely isotropic poroelastic material (Li, H.R. et al., 2012). Two theories validated the correctness of the provided theory, and the impacts of anisotropic permeability and transverse isotropic features on the threnody hydromechanical coupling behaviour of layered saturated media are detailed (Wang, L. et al., 2021).

An investigation on the creep and consolidation behavior of the layered saturated soils with overlying dry layers under the vertical load. With the aid of the Laplace–Hankel transform, typical viscoelastic models (e.g., the Kelvin, the Maxwell, or the standard linear solid model), and the correspondence principle, a semi-analytical solution is presented for this investigation (Niu, J. et al., 2020). Detailed comparisons between the present results with the published numerical and analytical results are given to confirm the solution, followed by an extensive parametric study examining the effect of types of viscoelastic models, thickness of the overlying layer, and viscosity (Wang, L. et al., 2022).

Cao Jiansheng, Zhang Wanjun, etc. further explored the water volume change law of weathered rock and soil by experimental research under the condition of fully considering the different fillings between rock blocks and fractures (Chen, S.L. et al., 2005), Fan Wen, Yan Furong and Lu Quanzhong further verified the adaptability of the binary medium model in loess through the comparative analysis of the calculation results of the binary medium model and the triaxial test results of fractured loess, and established the binary medium model for the mechanical properties of loess in the fractured zone (Qian H. et al., 1985), In 2013, Liu Enlong and Zhang Jianhai established a binary medium model of rock under cyclic load through triaxial experiment (Li, P. et al., 2016), The developed a triaxial apparatus having controlled suction and obtained the results of tests conducted on undisturbed samples. He reported successful measurement of the collapsible potential of loess sand (Wang, J. et al., 2020). Liu Enlong, Hu Zaiqiang, and Hou Feng further explored the applicability of the binary medium model in Loess by using the finite element me

---

## [Decision Letter · Decision Letter 1]

19 Jun 2025

Evaluation of Collapsible Deformation of foundation under rectangular load based on the improved binary medium model

PONE-D-25-11186R1

Dear Dr. Yosri,

We’re pleased to inform you that your manuscript has been judged scientifically suitable for publication and will be formally accepted for publication once it meets all outstanding technical requirements.

Kind regards,

Jianguo Wang, PhD

Academic Editor

PLOS ONE

Additional Editor Comments (optional):

Reviewers' comments:

Reviewer's Responses to Questions

**Comments to the Author**

1. If the authors have adequately addressed your comments raised in a previous round of review and you feel that this manuscript is now acceptable for publication, you may indicate that here to bypass the “Comments to the Author” section, enter your conflict of interest statement in the “Confidential to Editor” section, and submit your "Accept" recommendation.

Reviewer #1: All comments have been addressed

Reviewer #2: All comments have been addressed

2. Is the manuscript technically sound, and do the data support the conclusions?

Reviewer #1: Yes

Reviewer #2: Yes

3. Has the statistical analysis been performed appropriately and rigorously? 

Reviewer #1: Yes

Reviewer #2: Yes

4. Have the authors made all data underlying the findings in their manuscript fully available?

Reviewer #1: Yes

Reviewer #2: Yes

5. Is the manuscript presented in an intelligible fashion and written in standard English?

Reviewer #1: Yes

Reviewer #2: Yes

6. Review Comments to the Author

Reviewer #1: The comments and suggestions provided by the reviewer have been taken into consideration and incorporated into the manuscript. These improvements have resulted in a notable enhancement compared to the previous version of the article. As such, I agree to be accepted by the journal for publication

Reviewer #2: The authors have revised the paper according to the comments and addressed all the comments. It can be accepted as it is.

7. PLOS authors have the option to publish the peer review history of their article (what does this mean? ). If published, this will include your full peer review and any attached files.

**Do you want your identity to be public for this peer review?** For information about this choice, including consent withdrawal, please see our Privacy Policy .

Reviewer #1: No

Reviewer #2: No

---

## [Editor Report · Acceptance letter]

PONE-D-25-11186R1

PLOS ONE

Dear Dr. Yosri,

I'm pleased to inform you that your manuscript has been deemed suitable for publication in PLOS ONE. Congratulations! Your manuscript is now being handed over to our production team.

Kind regards,

on behalf of

Dr. Jianguo Wang

Academic Editor

PLOS ONE